# Nutritional Influences on Locomotive Syndrome

**DOI:** 10.3390/jcm11030610

**Published:** 2022-01-26

**Authors:** Sadayuki Ito, Hiroaki Nakashima, Kei Ando, Masaaki Machino, Taisuke Seki, Shinya Ishizuka, Yasuhiko Takegami, Kenji Wakai, Yukiharu Hasegawa, Shiro Imagama

**Affiliations:** 1Department of Orthopedic Surgery, Nagoya University Graduate School of Medicine, Nagoya 466-8560, Japan; sadaito@med.nagoya-u.ac.jp (S.I.); andokei@med.nagoya-u.ac.jp (K.A.); masaaki_machino_5445_2@yahoo.co.jp (M.M.); taiseki@med.nagoya-u.ac.jp (T.S.); shinyai@med.nagoya-u.ac.jp (S.I.); takegami@med.nagoya-u.ac.jp (Y.T.); imagama@med.nagoya-u.ac.jp (S.I.); 2Department of Preventive Medicine, Nagoya University Graduate School of Medicine, Nagoya 466-8550, Japan; wakai@med.nagoya-u.ac.jp; 3Department of Rehabilitation, Kansai University of Welfare Science, Osaka 582-0026, Japan; hasegawa@tamateyama.ac.jp

**Keywords:** locomotive syndrome, nutrition, health checkup data

## Abstract

Healthy dietary habits are important to prevent locomotive syndrome (LS). We investigated the relationship between LS and nutritional intake using community health checkup data. We included 368 participants who underwent LS staging, blood sampling, and nutritional intake assessments. Participants (163 adults < 65: 205 older adults ≥ 65) were divided into normal (N; LS stage 0) and LS (L; LS stage 1–2) groups, and blood sample data and nutritional intake were compared between groups. Among adults (N group, 71; L group, 92), low-density lipoprotein cholesterol (LDL-C) was significantly lower, and Vitamin B_1_ intake was significantly higher in the L than in the N group; LDL-C, *p* = 0.033; Vitamin B1, 0.029. Among older adults (N group, 85; L group, 120), hemoglobin (Hb), albumin, and calcium levels were significantly lower, and sodium, monounsaturated fatty acids (MUFA), and n-6 polyunsaturated fatty acids (n-6 PUFA) were significantly higher in the L than the N group; Hb, *p* = 0.036; albumin, 0.030; calcium, 0.025; sodium; 0.029; MUFA; 0.047, n-6 PUFA; 0.0233). Logistic regression analysis indicated that sodium was the risk factor for the L group (exp (B) 1.001, 95% CI: 1–1.001, *p* = 0.032). In conclusion, salt intake was associated with LS.

## 1. Introduction

Most developed countries are facing an aging society [1], and the number of people who need support and care in daily life due to musculoskeletal disorders is increasing [2]. Locomotive syndrome (LS) has been proposed by the Japanese Orthopaedic Association (JOA) as an umbrella term to refer to the condition of reduced mobility due to musculoskeletal disorders [2]. Details of the LS can be viewed at the following website; https://locomo-joa.jp/assets/pdf/index_english.pdf (accessed on 23 January 2022). Locomotive syndrome is defined as a condition of reduced mobility due to the impairment of locomotive organs. LS is assessed by evaluating the degree of motor function via the two-step test, stand-up test, and 25-question Geriatric Locomotive Function Scale (GLFS-25). LS has received worldwide attention for assessing motor function in musculoskeletal diseases [3]. LS is associated with a significantly lower quality of life (QOL) [4] and shorter life expectancy. Prevention of LS has long been advocated for maintaining and improving physical function in middle-aged and older adults [5]. In recent years, there have been reports using LS as an index of postoperative outcome. [6,7].

Visceral diseases and genetic factors can also contribute to LS. Variable factors, such as habitual inactivity, sedentary lifestyle, and inadequate nutrition contribute to the progression of LS [8]. Therefore, the JOA proposed an exercise for LS called locomotion training (LT), which consists of standing on one leg with the eyes open and performing squats, heel raises, and front lunges [9].

In addition to reduced mobility, malnutrition is a common condition in older adults and is associated with morbidity, mortality, and reduced quality of life [10,11]. Malnutrition also affects musculoskeletal health and has been associated with chronic musculoskeletal pain in older adults [12]. Malnutrition in older adults can progress through a variety of mechanisms. Immobility due to musculoskeletal conditions has been associated with sarcopenia and decreased oral intake [13]. Impaired mobility and malnutrition, two conditions prevalent among older individuals, may be correlated and may potentiate each other, resulting in poor health-related outcomes [10]. Muscle strength is the direct key that links impaired mobility to malnutrition. Decreased muscle strength has been reported to be an indicator of impaired mobility [14], and decreased muscle mass has been reported to be a consequence of malnutrition [15].

There are no reports directly examining the relationship between LS and dietary habits. This study aimed to examine the relationship between LS and nutritional intake and identify the nutrients and dietary habits that influence LS.

## 2. Materials and Methods

### 2.1. Study Participants

The individuals surveyed were volunteers who underwent a municipal-supported health checkup in Yakumo in 2016 and 2017. Data from 2016 were used for those who participated in two consecutive years (2016 and 2017). Yakumo has a population of approximately 17,000, of whom 28% are >65 years old. More people in this town engage in agriculture and fishing than those in urban areas. Yakumo has conducted annual health checkups since 1982. Physical examinations included voluntary orthopedic and physical function tests and internal examinations. Psychological examinations and a health-related QOL survey (SF-36) were also conducted [16]. This study included all participants who completed an assessment of the LS risk stage, bioelectrical impedance analysis (BIA), fasting blood samples, and nutritional intake status. The exclusion criteria were as follows: history of spine or joint surgery, severe knee injury, severe hip osteoarthritis, history of hip or spine fractures, neuropathy, severe mental illness, diabetes that was diagnosed and treated by a physician, kidney or heart disease, nonfasting, severe impairment of walking or standing, and impairment of the central or peripheral nervous system.

Of the 758 participants who underwent health checks, 368 (154 men; 214 women) met the inclusion criteria. The research protocol was approved by the Human Research Ethics Committee and the university’s institutional review board (No. 2014-0207). All participants provided written informed consent before participation. The research procedure was conducted in accordance with the principles of the Declaration of Helsinki.

### 2.2. Examination of Motor Function

Grip strength in the standing position was measured once in each hand using a handgrip dynamometer (Toei Light Co., Ltd., Saitama, Japan). The mean value was used for analysis [17]. Participants walked a 10 m straight course once at their fastest pace, and the time taken to complete the course was recorded as the 10 m walking time [18].

### 2.3. LS Stage Tests

The JOA proposes three tests to assess the risk of LS by evaluating the degree of motor function: the two-step test, stand-up test, and 25-question Geriatric Locomotive Function Scale (GLFS-25) [2]. LS is classified into stages 1 and 2. These stages are defined as follows: stage 1 indicates that motor function is beginning to decline and stage 2 indicates that motor function is progressing toward decline. Three tests were conducted according to the JOA guidelines [2].

The standing test assessed the ability to stand on one or both feet from stools of 40, 30, 20, and 10 cm height. The difficulty rating from easy to difficult was based on standing on both legs from stools of 40, 30, 20, and 10 cm, followed by standing on one leg from stools of 40, 30, 20, and 10 cm. The test results were expressed as the minimum height of the stool that the participant could stand on.

In the two-step test, the physical therapist measured the length of two steps from the starting line to the tip of the toes. The score was calculated by normalizing the maximum length of the two steps by the height.

The GLFS-25 is a comprehensive self-report survey that refers to the previous month [19]. The method consists of four questions on pain, 16 questions on activities of daily living (ADL), three questions on social functioning, and two questions on mental status. Each item was rated from no disability (0 points) to severe disability (4 points).

LS 0, 1, and 2 were defined as follows:

LS 0

The participant was categorized as Stage 0 if all of the three following conditions were met:Stand-up test, ability in one-leg standing from a 40 cm-high seat (both legs).Two-step test, >1.3.25-question GLFS score, <7.

LS1

The participant was categorized as Stage 1 if any of the three following conditions were met:Stand-up test, difficulty in one-leg standing from a 40 cm-high seat (either leg).Two-step test, <1.3.25-question GLFS score, ≥7.

LS2

The participant was categorized as Stage 2 if any of the three following conditions were met:Stand-up test, difficulty in standing from a 20 cm-high seat using both legs.Two-step test, <1.1.25-question GLFS score, ≥16 [20].

We divided the participants into two groups: the normal (N) group (LS 0) and the locomotive syndrome (L) group (LS1,2).

### 2.4. Bioelectrical Impedance Analysis (BIA)

BIA was used to analyze the participants’ body composition. The participants underwent the BIA on an empty stomach. The conditions of BIA measurement, such as consumption of food and beverages, were similar to those reported earlier [21]. Anthropometric data, including height, weight, body mass index (BMI), body fat percentage (BFP), and appendicular skeletal muscle mass index (SMI), were measured using the BIA. The Inbody 770 BIA device (Inbody Co., Ltd., Seoul, Korea), which can differentiate between tissues (such as fat, muscle, and bone) based on their electrical impedance, was used for the participants’ body composition [16]. The accuracy of this device has been reported previously [22]. Participants grasped the handles of the analyzer, which have embedded electrodes, and stood on the platform with the soles of their feet in contact with the electrodes. There were two electrodes for each foot and hand. The BMI was calculated using the following formula: weight (kg)/height^2^ (m^2^). The muscle mass of each limb was automatically calculated by BIA using the Inbody 770 BIA device. The SMI was calculated using the following formula: SMI = appendicular skeletal muscle mass (kg)/height^2^ (m^2^) [23].

### 2.5. Blood Sample Assessment

At the checkup, fasting blood samples were collected by venipuncture and centrifuged within 1 h of collection. Serum samples were stored at −80 °C until measurements were taken. Routine biochemical analysis was performed in the laboratory of Yakumo Town Hospital [24]. The following items were investigated; white blood cell, hemoglobin, platelet, HbA1c, total protein, serum albumin, alkaline phosphatase (ALP), aspartate transaminase (AST), alanine aminotransferase (ALT), γ-glutamyltranspeptidase, total cholesterol, triglyceride, high-density lipoprotein cholesterol (HDL-C), low-density lipoprotein cholesterol (LDL-C), blood urea nitrogen, creatinine, uric acid, calcium, C-reactive protein.

### 2.6. Lifestyle Habits

Trained nurses administered a questionnaire on health and daily lifestyle habits, including smoking (current smoker, ex-smoker, or nonsmoker), alcohol consumption (regular drinkers, ex-drinkers, or nondrinkers), nutritional intake status, menopausal status (yes or no), and history of major illness. Anthropometric indices (height and weight) and blood pressure were measured during the health examination. Body mass index (BMI) was calculated as body weight (kg) divided by height (m) squared.

Hypertension was defined as systolic blood pressure of 140 mmHg or higher or diastolic blood pressure of 90 mmHg or higher (based on the Japanese Society of Hypertension guidelines) [25] or the use of antihypertensive medications. Diabetes mellitus was defined as a fasting blood glucose level of 126 mg/dL or higher or a glycated hemoglobin (HbA1c) level of 6.5% or higher or use of antidiabetic drugs. The Japanese Diabetes Society (JDS) HbA1c value (%) was converted to the equivalent National Glycohemoglobin Standardization Program (NGSP) value (%) using the formula HbA1c (NGSP) = HbA1c (JDS) + 0.4% [26]. Dyslipidemia was defined as a triglyceride level of ≥150 mg/dL, high-density lipoprotein cholesterol (HDL-C) level of < 40 mg/dL, or a low-density lipoprotein cholesterol (LDL-C) level of ≥140 mg/dL (based on the Japan Atherosclerosis Society guidelines) [27], or use of antidyslipidemic drugs.

### 2.7. Nutritional Intake Status

Dietary information was obtained using a validated food frequency questionnaire (FFQ), which asked about the intake of 188 food and beverage items, excluding supplements. Food and beverage items were grouped into nine categories ranging in frequency from “rarely” to “7 or more times a day” (or “10 or more drinks a day” for beverages). The question asked about usual consumption of the listed foods over the past year. The food list was initially developed from the 1989–1991 weighed food record according to contribution rates based on absolute values of energy and intake of 14 target nutrients, and was used in a prospective Japan Health Center-based study 8–12 [28] modified for middle-aged and older residents in a wide range of regions in Japan. In doing so, we took into account 17 additional nutrients, such as dietary fiber and folate, changes in foods contributing to the absolute nutrient intake due to updates to the Standard Tables of Food Composition in Japan [29], and regional and generational changes in diet in the current cohort. Energy and nutrient intakes were calculated by summing the product of the frequency of eating, portion size, energy, and nutrient content of each food item, referring to the Fifth Revised and Expanded Standard Tables of Food Composition in Japan [30]. Nutrients included protein, fat, carbohydrate, minerals (sodium, potassium, calcium, and iron), vitamins (carotene, Vitamins A, D, E, B_1_, B_2_, folate, and C), and total dietary fiber (TDF) (soluble DF and insoluble DF). Fat was divided into saturated fatty acids, monounsaturated fatty acids (MUFA), n-6 and n-3 polyunsaturated fatty acids (PUFAs), and n-3 highly unsaturated fatty acids (n-3 HUFAs, including eicosapentaenoic acid (20:5), docosapentaenoic acid (22:5), docosahexaenoic acid (22:6)), and cholesterol [31].

### 2.8. Statistical Analyses

We divided all participants into adults (< 65 years old) and older adults (≥ 65 years old). Continuous variables were expressed as mean ± standard deviation (SD). We compared continuous variables of the L group to those of the N group using the Student’s *t*-test and categorical variables using the chi-square test. These analyses were conducted for total adult and older adult, respectively. Statistical significance was set at *p* < 0.05. All the parameters listed in the Appendix A were examined. In addition, the table shows the parameters that showed significant differences and their related parameters.

Logistic regression analysis was performed on each adult and older adult to evaluate the important risk factors in the L group. In the logistic regression analyses, we defined that the dependent variable was the group and that the covariables were the parameters that showed significant differences in the Student *t*-test and the chi-square test comparing the L Group and N Group. All statistical analyses were performed using SPSS Statistics v.28.0 software for Mac (IBM Corp., Armonk, NY, USA). Statistical significance was set at *p* < 0.05.

## 3. Results

Participant characteristics are shown in Table 1 and Appendix A. There were 154 men and 214 women, with an average age of 63.8 ± 10.5 years. There were 156 (76 male; 80 female) and 212 (78 male; 134 female) participants in the N and L groups, respectively.

### 3.1. Adult Participants

The average age of the adult group was 54.3 ± 7.3 years. In total, 71 (28 male; 43 female) and 92 (24 male; 68 female) participants were included in the N and L groups, respectively (Table 2, Appendix A). The body fat percentage (BFP) was significantly higher and the grip strength and gait speed were significantly lower in the L than in the N group (BFP; N: 28.2 ± 5.1, L: 31.0 ± 6.8, *p* = 0.005, grip strength; N: 29.6 ± 9.8, L: 25.1 ± 8.7, *p* = 0.002, gait speed; N: 2.4 ± 0.3, L: 2.3 ± 0.5, *p* = 0.04).

Analyses of the laboratory data and nutrient intakes indicated that LDL-C was significantly lower and Vitamin B_1_ was significantly higher in the L than in the N group (LDL-C; N: 132.1 ± 32.4, L: 121.8 ± 28.5, *p* = 0.033, Vitamin B_1_; N: 0.66 ± 0.07, L: 0.69 ± 0.09, *p* = 0.029). There were no significant between-group differences in the other laboratory data, nutrient intake, SMI, or history of metabolic diseases (Table 2, Appendix A).

As significant differences were observed among several factors, they were examined as covariates for risk factors of the L group in the logistic regression analysis. BFP and LDL-C were risk factors (BFP; exp (B) 1.068, 95% CI: 1.003–1.127, *p* = 0.038, LDL-C; exp (B) 0.988, 95% CI: 0.977–1, *p* = 0.042) (Table 3).

### 3.2. Older Adult Participants

The average age of older adult participants was 71.3 ± 5.3 years. The N and L groups contained 85 (48 male: 37 female) and 120 (54 male: 66 female) participants, respectively (Table 4, Appendix A). The average age of participants was significantly higher in the L than in the N group (N: 70.2 ± 4.7; L: 72.2 ± 5.6, *p* = 0.008). The BFP was significantly higher and the grip strength was significantly lower in the L than in the N group (BFP; N: 26.7 ± 6.3, L: 29.7 ± 6.9, *p* = 0.002, grip strength; N: 29.0 ± 8.0, L: 25.1 ± 8.0, *p* = 0.001).

Analyses of laboratory data indicated that Hb, albumin, and calcium levels were significantly lower in the L than in the N group (Hb: 13.8 ± 1.0, L: 13.4 ± 1.1, *p* = 0.036, albumin; N: 4.4 ± 0.2, L: 4.3 ± 0.2, *p* = 0.030, calcium; N: 9.2 ± 0.3, L: 9.1 ± 0.3, *p* = 0.025). Regarding nutrient intake, sodium, MUFA, and n-6 PUFA were significantly higher in the L than in the N group (sodium: N: 1961.1 ± 586.7, L: 2183.1 ± 789.0, *p* = 0.029; MUFA: N: 15.8 ± 3.8, L: 17.2 ± 5.4, *p* = 0.047, n-6 PUFA; N: 10,898.3 ± 2808.4, L: 12,149.4 ± 4807.4, *p* = 0.0233). The prevalence of hypertension was higher in the L group than in the N group (N: 42.3%, L: 60.8%, *p* = 0.007). There were no significant between-group differences in gait speed, SMI, other laboratory data, nutrient intake, or history of metabolic diseases between the N and L groups (Table 4).

Factors for which significant between-group differences were observed were examined as covariates for the risk factors of the L group in the logistic regression analysis. BFP, sodium, and hypertension were risk factors (exp (B) 1.073, 95% CI: 1.017–1.132, *p* = 0.011, sodium; exp (B) 1.001, 95% CI: 1–1.001, *p* = 0.032, hypertension; exp (B) 2.288, 95% CI: 1.186–4.412, *p* = 0.014) (Table 5).

## 4. Discussion

There have been several reports on the relationship between motor function and muscle and nutrition [10,11]. However, to our knowledge, no study has directly examined the relationship between nutritional intake status and LS. In this study, salt, MUFA, and n-6 PUFA intakes were significantly higher in group L among older adults. Multivariate analysis indicated that higher salt intake was particularly associated with LS.

In an aging society, the rate of LS is increasing, and the number of people who have difficulty leading independent daily lives is increasing [4]. Therefore, preventing LS plays a vital role in preventing the need for nursing care. In this study, a significant decrease in grip strength was observed in the L group in both adults and older adults. This suggests that a decline in muscle strength accompanies the functional decline in LS and that the influence of muscle strength on LS is significant and closely related to sarcopenia. Exercise and nutrition are important factors in preventing sarcopenia [32] and are thought to be equally important in influencing LS. Exercise therapy has been advocated for LS prevention [9]. However, there have been no detailed studies on the effects of nutrition on LS. Thus, there is no specific diet therapy for LS prevention.

BFP and LDL-C levels were lower in group L among adults, according to the multivariate analysis. Higher body fat may reflect lower physical activity, such as lack of exercise, in group L. High LDL cholesterol is associated with a higher percentage of fat than muscle [33]. However, in this study, LDL-C levels of both groups were within the normal range, so the difference was not clinically significant.

Vitamin B_1_ intake was lower in group L among adults, according to the univariate analysis. Vitamin B_1_ is active in glucose metabolism, and its deficiency can lead to poor conversion of sugar to energy and increased fatigue [34]. In this study, the intake of Vitamin B_1_ was higher in group L, but both groups had deficient Vitamin B_1_ intake compared to the daily recommended amount [35], and the difference in intake was small and not significant for LS.

Among older adults, group L had higher salt intake and hypertension, according to the multivariate analysis. Excessive salt intake is one of the causes of various diseases, such as hypertension [36,37], where sodium in the skeletal muscle accumulates more in older than in younger people, and patients with refractory hypertension have increased tissue sodium (Na^+^) content compared to normotensive controls [38]. Moreover, the sodium-potassium-chloride symporter 1 (NKCC1) is highly expressed in mammalian skeletal muscles. The physiological function of NKCC1 in myogenesis remains unclear. However, NKCC1 protein levels increase skeletal myoblast differentiation, and NKCC1 inhibitors markedly suppress skeletal myoblast differentiation [39]. It has also been reported that excess sodium leads to downregulation of NKCC expression [40]. Further, the risk of sarcopenia, which is associated with decreased muscle mass and strength, is related to the amount of salt intake because Na^+^ is stored in tissues, and NKCC1 is involved in muscle hypertrophy and suppression [41]. Here, salt intake was higher in the L group.

Regarding hypertension and muscle, hypertension reduces muscle blood flow [42] and correlates with low muscle mass [43]. Arterial stiffness and age-related hormonal decline are risk factors for hypertension and muscle loss [44]. Similar to these observations, this study also detected an association between hypertension and LS. Thus, higher salt intake and hypertension were associated with muscle metabolism and sarcopenia, which may be associated with the locomotive syndrome. In this study, we found that the locomotive syndrome group had weaker grip strength, which may be due to muscle weakness and sarcopenia. Therefore, it can be said that factors related to muscle mass loss are closely related to the locomotive syndrome. 

In contrast, even after multivariate analysis of confounding factors involved in locomotive syndrome, including grip strength, higher salt intake was significantly associated with locomotive syndrome. Higher salt intake may also be related to motor nerves, sensory nerves that control balance, and other factors, in addition to its effect on muscle mass. There are many unclear points regarding the relationship between higher salt intake and nonmuscle factors; thus, further basic research is needed in the future.

Among older adults, there were no significant differences except for salt intake and hypertension in multivariate analysis. However, there were significant differences in MUFA and PUFA intake, hemoglobin, serum albumin, and calcium in the univariate analysis. Although there was no significant multivariate difference between n-6 and n-3 PUFA levels and locomotive syndrome, there may be a small relationship with locomotive function. In recent years, several studies have reported that n-6 and n-3 PUFAs have opposite effects on insulin resistance (IR) and body homeostasis [45]. It is postulated that n-3 PUFAs attenuate the development of IR by reducing inflammation, whereas n-6 PUFAs promote IR [46]. More recent evidence indicates that n-6 PUFAs play a key role in the inflammatory process and are associated with various metabolic diseases [47]. Therefore, it seems reasonable to control the dietary ratios of n-6/n-3 PUFAs to ameliorate obesity-related IR, which is beneficial for protection against chronic and metabolic diseases. A study of rats fed a high-fat diet demonstrated that the accumulation of fatty acids in various lipid fractions increased in the gastrocnemius red muscle when there was a shift in the n-6/n-3 PUFA balance in favor of n-6 PUFAs. The increases in n-6 PUFAs are associated with fat accumulation in muscle [48]. A high percentage of n-6 PUFAs was also associated with LS in this study.

This study had several limitations. First, the participants were middle-aged and older adults who lived in a relatively rural area and were employed in agriculture or fishing. Thus, the differences in lifestyles between these participants and people in an urban environment may limit the generalizability of our results. Second, the participants attended annual health examinations, suggesting that they may be more health-conscious than other people. Third, this was a cross-sectional, single-center study. In the future, longitudinal and multicenter collaborative research is needed to verify our findings. Fourth, BIA is not the gold standard assessment for body composition (the 4-compartment model or DXA are better).

## 5. Conclusions

In conclusion, this study examined the relationship between LS and nutrition. Salt intake and hypertension were associated with locomotive syndrome. Thus, we suggest that limiting salt intake may effectively prevent LS.

## Figures and Tables

**Table 1 jcm-11-00610-t001:** The comparison of each parameter between nonolder adult and older adult participants.

	All (*n* = 368)	Adult (*n* = 163)	Older Adult (*n* = 205)	*p*
Male/Female	154/214	52/111	102/103	0.001 *
Age (yrs)	63.8 ± 10.5	54.3 ± 7.3	71.3 ± 5.3	<0.001 *
Height (cm)	158.1 ± 8.1	159.4 ± 7.8	157 ± 8.2	0.006 *
Weight (kg)	59.2 ± 11.5	59.6 ± 12.3	58.9 ± 10.8	0.553
BMI (kg/m^2^)	23.5 ± 3.5	23.3 ± 3.6	23.7 ± 3.4	0.231
BFP (%)	29.1 ± 6.6	29.8 ± 6.3	28.5 ± 6.8	0.057
SMI (kg/m^2^)	6.70 ± 1.03	6.65 ± 1.07	6.73 ± 1.00	0.477
Grip strength (kg)	26.9 ± 8.8	27.1 ± 9.4	26.7 ± 8.2	0.730
N/L	156/212	71/92	85/120	0.383
Hypertension (y/n)	140/228	31/132	109/96	<0.001 *
Diabetes (y/n)	24/344	3/160	21/184	0.001 *
Hypertension (y/n)	140/228	31/132	109/96	<0.001 *
Laboratory data				
Hemoglobin (g/dL)	13.5 ± 1.2	13.4 ± 1.3	13.6 ± 1.1	0.175
Serum Albumin (g/dL)	4.4 ± 0.2	4.4 ± 0.2	4.3 ± 0.2	0.183
Total-cholesterol (mg/dL)	207.2 ± 33.2	212.6 ± 31.8	202.9 ± 33.7	0.005 *
Triglyceride (mg/dL)	111.2 ± 69.5	103.7 ± 59.9	117.1 ± 75.8	0.067
HDL-C (mg/dL)	61.5 ± 14.9	62.9 ± 14.5	60.4 ± 15.1	0.103
LDL-C (mg/dL)	120.5 ± 30.7	126.3 ± 30.6	115.8 ± 30	0.001 *
Calcium (mg/dL)	9.2 ± 0.3	9.2 ± 0.3	9.2 ± 0.3	0.726
Nutritional intake				
Energy (kcal/day)	1644.1 ± 389.5	1591.1 ± 351.2	1686.3 ± 413.4	0.020 *
Protein (g/day)	53.3 ± 13.7	51.5 ± 10.8	54.8 ± 15.6	0.022 *
Fat (g/day)	44.6 ± 13.5	42.9 ± 12.2	46 ± 14.3	0.031 *
Carbohydrate (g/day)	229.1 ± 69.5	221.1 ± 67.6	235.4 ± 70.6	0.050
Sodium (mg/day)	1972.3 ± 689.4	1822.9 ± 620.8	2091.1 ± 719	<0.001 *
Calcium (mg/day)	543.9 ± 191.6	502.7 ± 150.7	576.7 ± 213.5	<0.001 *
SFA (g/day)	11.6 ± 2.9	11.1 ± 2.4	11.9 ± 3.2	0.011 *
MUFAd (g/day)	16.4 ± 4.8	16.1 ± 4.8	16.6 ± 4.8	0.293
PUFA (g/day)	13.2 ± 4.5	12.5 ± 4.0	13.8 ± 4.9	0.009 *
Cholesterol (mg/day)	241.6 ± 79.1	238.9 ± 78.4	243.7 ± 79.8	0.560
n-3 PUFA (g/day)	2299.2 ± 750.9	2195 ± 764.7	2382 ± 731	0.017 *
n-6 PUFA (g/day)	11,206.8 ± 3901.2	10,673.7 ± 3523.9	11,630.7 ± 4136.5	0.019 *
Energy from alcohol (kcal/day)	47.2 ± 96.7	49 ± 101.5	45.7 ± 92.9	0.742
n-3 HUFA (g/day)	754.6 ± 403.1	682.2 ± 344.5	812.1 ± 436.5	0.002 *

Values are expressed as means ± standard deviations; BMI: body mass index, BFP: body fat percentage, SMI: skeletal muscle mass index, y/n: yes/no; SMI: skeletal muscle mass index, N/L: normal group/locomotive syndrome group, y/n: yes/no; HDL-C: high-density lipoprotein cholesterol, LDL-C: low-density lipoprotein cholesterol; SFA: saturated fatty acid, MUFA: monounsaturated fatty acid, PUFA: polyunsaturated fatty acid, HUFA: highly unsaturated fatty acid. All analyses are comparisons between adults and older adults. The comparison of male/female was conducted by a chi-square test, and the comparisons of the others were conducted by a Student *t*-test. *: *p* < 0.05.

**Table 2 jcm-11-00610-t002:** The comparison of each parameter between the N and L groups in adult participants.

	Adult (*n* = 163)	N (*n* = 71)	L (*n* = 92)	*p*
Male/Female	52/111	28/43	24/68	0.090
Age (yrs)	54.3 ± 7.3	53.2 ± 8	55.1 ± 6.7	0.094
BMI (kg/m^2^)	23.3 ± 3.6	22.8 ± 3.3	23.7 ± 3.8	0.164
BFP (%)	29.8 ± 6.3	28.2 ± 5.1	31.0 ± 6.8	0.005 *
SMI (kg/m^2^)	6.65 ± 1.07	6.72 ± 1.04	6.60 ± 1.09	0.489
Grip strength (kg)	27.1 ± 9.4	29.6 ± 9.8	25.1 ± 8.7	0.002 *
Hypertension (y/n)	31/132	9/62	22/70	0.075
Diabetes (y/n)	3/160	3/68	0/92	0.081
Hyperlipidemia (y/n)	27/136	11/60	16/76	0.833
Laboratory data				
Hemoglobin (g/dL)	13.4 ± 1.3	13.5 ± 1.5	13.3 ± 1.1	0.268
Serum Albumin (g/dL)	4.4 ± 0.2	4.4 ± 0.2	4.4 ± 0.2	0.139
Total-cholesterol (mg/dL)	212.6 ± 31.8	216.2 ± 36.9	209.8 ± 27.2	0.205
Triglyceride (mg/dL)	103.7 ± 59.9	104.6 ± 69.8	103.1 ± 51.5	0.879
HDL-C (mg/dL)	62.9 ± 14.5	61.8 ± 14.7	63.8 ± 14.4	0.397
LDL-C (mg/dL)	126.3 ± 30.6	132.1 ± 32.4	121.8 ± 28.5	0.033
Calcium (mg/dL)	9.2 ± 0.3	9.2 ± 0.3	9.2 ± 0.3	0.962
Nutritional intake				
energy (kcal/day)	1591.1 ± 351.2	1617.7 ± 360.7	1570.6 ± 344.4	0.397
protein (g/day)	51.5 ± 10.8	51.3 ± 9	51.7 ± 12	0.808
fat (g/day)	42.9 ± 12.2	42.0 ± 9.2	43.6 ± 14.1	0.408
carbohydrate (g/day)	221.1 ± 67.6	228 ± 71.9	215.7 ± 63.9	0.251
Sodium (mg/day)	1822.9 ± 620.8	1879.4 ± 612.3	1779.3 ± 627.2	0.309
Calcium (mg/day)	502.7 ± 150.7	507.8 ± 152.7	498.8 ± 149.8	0.708
SFA (g/day)	11.1 ± 2.4	11.2 ± 2.6	11.1 ± 2.2	0.903
MUFAd (g/day)	16.1 ± 4.8	15.5 ± 3.3	16.5 ± 5.7	0.170
PUFA (g/day)	12.5 ± 4	12.3 ± 2.9	12.7 ± 4.6	0.571
Cholesterol (mg/day)	238.9 ± 78.4	222.6 ± 50.9	251.4 ± 92.7	0.020 *
n-3 PUFA (g/day)	2195 ± 764.7	2100.3 ± 481.5	2268 ± 922.2	0.166
n-6 PUFA (g/day)	10,673.7 ± 3523.9	10,579.5 ± 2506.8	10,746.4 ± 4154.5	0.765
Energy from alcohol (kcal/day)	49.0 ± 101.5	39.2 ± 67.3	56.6 ± 121.4	0.280
n-3 HUFA (g/day)	682.2 ± 344.5	640 ± 274.9	714.8 ± 388.1	0.170

Values are expressed as means ± standard deviations. BMI: body mass index, BFP: body fat percentage, SMI: skeletal muscle mass index, y/n: yes/no, HDL-C: high-density lipoprotein cholesterol, LDL-C: low-density lipoprotein cholesterol, SFA: saturated fatty acid, MUFA: monounsaturated fatty acid, PUFA: polyunsaturated fatty acid, HUFA: highly unsaturated fatty acid. All analyses are comparisons between the N and L group. The comparisons of male/female, hypertension, diabetes, and hyperlipidemia were conducted by a chi-square test, and the comparison of the others were conducted by a Student *t*-test. *: *p* < 0.05. There were significant differences in BFP, grip strength, gait speed, LDL-C, and Vitamin B_1_ between the N and L groups.

**Table 3 jcm-11-00610-t003:** Logistic regression analysis for risk factors of the locomotive syndrome (L group) in adult participants.

	B	SE	Wald	df	*p*	Exp(B)	95% CI
BFP	0.061	0.030	4.287	1	0.038 *	1.063	1.003–1.127
Grip strength	−0.036	0.019	3.624	1	0.057	0.964	0.929–1.001
Gait speed	−0.686	0.377	3.308	1	0.069	0.504	0.240–1.055
LDL-C	−0.012	0.006	4.128	1	0.042 *	0.988	0.977–1.000
Vitamin B1	3.799	2.140	3.150	1	0.076	44.639	0.673–2961.192

BFP: body fat percentage, LDL-C: low-density lipoprotein cholesterol. *: *p* < 0.05. Covariates: BFP, grip strength, gait speed, LDL-C, Vitamin B_1_. There were significant differences in BFP and LDL-C.

**Table 4 jcm-11-00610-t004:** The comparison of each parameter between the N and L groups in older adult participants.

	Older Adult (*n* = 205)	N (*n* = 85)	L (*n* = 120)	*p*
male/female	102/103	48/37	54/66	0.120
Age (yrs)	71.3 ± 5.3	70.2 ± 4.7	72.2 ± 5.6	0.008 *
BMI (kg/m^2^)	23.7 ± 3.4	23.2 ± 3.4	24.1 ± 3.3	0.061
BFP (%)	28.5 ± 6.8	26.7 ± 6.3	29.7 ± 6.9	0.002 *
SMI (kg/m^2^)	6.73 ± 1.00	6.86 ± 1.06	6.64 ± 0.95	0.140
grip strength (kg)	26.7 ± 8.2	29.0 ± 8.0	25.1 ± 8.0	0.001 *
Hypertension (y/n)	109/96	36/49	73/47	0.007 *
Diabetes (y/n)	21/184	6/79	15/105	0.157
Hyperlipidemia (y/n)	81/124	36/49	45/75	0.305
Laboratory data				
Hemoglobin (g/dL)	13.6 ± 1.1	13.8 ± 1.0	13.4 ± 1.1	0.036 *
Serum Albumin (g/dL)	4.3 ± 0.2	4.4 ± 0.2	4.3 ± 0.2	0.030 *
Total-cholesterol (mg/dL)	202.9 ± 33.7	205.3 ± 35	201.2 ± 32.8	0.399
Triglyceride (mg/dL)	117.1 ± 75.8	112.6 ± 59.9	120.3 ± 85.4	0.479
HDL-C (mg/dL)	60.4 ± 15.1	61.1 ± 15.3	59.8 ± 15.0	0.55
LDL-C (mg/dL)	115.8 ± 30	118.4 ± 28.2	114 ± 31.3	0.305
Calcium (mg/dL)	9.2 ± 0.3	9.2 ± 0.3	9.1 ± 0.3	0.025 *
Nutritional intake				
energy (kcal/day)	1686.3 ± 413.4	1638.8 ± 348.7	1719.9 ± 452	0.167
protein (g/day)	54.8 ± 15.6	52.9 ± 11.6	56.2 ± 17.8	0.132
fat (g/day)	46 ± 14.3	44 ± 12.1	47.4 ± 15.6	0.090
carbohydrate (g/day)	235.4 ± 70.6	227.2 ± 62	241.2 ± 75.8	0.162
Sodium (mg/day)	2091.1 ± 719	1961.1 ± 586.7	2183.1 ± 789	0.029 *
Calcium (mg/day)	576.7 ± 213.5	565.9 ± 182.2	584.4 ± 233.7	0.543
SFA (g/day)	11.9 ± 3.2	11.8 ± 3.1	12.0 ± 3.3	0.720
MUFAd (g/day)	16.6 ± 4.8	15.8 ± 3.8	17.2 ± 5.4	0.047 *
PUFA (g/day)	13.8 ± 4.9	13.0 ± 3.3	14.3 ± 5.7	0.063
Cholesterol (mg/day)	243.7 ± 79.8	234.4 ± 64.3	250.3 ± 88.9	0.161
n-3 PUFA (g/day)	2382 ± 731	2271.3 ± 518.1	2460.4 ± 843.6	0.068
n-6 PUFA (g/day)	11,630.7 ± 4136.5	10,898.3 ± 2808.4	12,149.4 ± 4807.4	0.033 *
Energy from alcohol (kcal/day)	45.7 ± 92.9	51.9 ± 87.2	41.3 ± 96.9	0.426
n-3 HUFA (g/day)	812.1 ± 436.5	781 ± 274.1	834.1 ± 522	0.393

Values are expressed as means ± standard deviations. BMI: body mass index, BFP: body fat percentage, SMI: skeletal muscle mass index, y/n: yes/no, HDL-C: high-density lipoprotein cholesterol, LDL-C: low-density lipoprotein cholesterol, SFA: saturated fatty acid, MUFA: monounsaturated fatty acid, PUFA: polyunsaturated fatty acid, HUFA: highly unsaturated fatty acid. All analyses are comparisons between N and L group. The comparisons of male/female, hypertension, diabetes, and hyperlipidemia were conducted by a chi-square test, and the others were conducted by a Student *t*-test. *: *p* < 0.05. There were significant differences in age, BFP, grip strength, hemoglobin, albumin, calcium, sodium, MUFA, n-6 PUFA, and hypertension between the N and L groups.

**Table 5 jcm-11-00610-t005:** Logistic regression analysis for risk factors of the locomotive syndrome (L group) in older adult participants.

	B	SE	Wald	df	*p*	Exp(B)	95% CI
Age (yrs)	0.057	0.035	2.633	1	0.105	1.059	0.988–1.135
BFP (%)	0.070	0.027	6.534	1	0.011 *	1.073	1.017–1.132
Grip strength (kg)	−0.036	0.023	2.514	1	0.113	0.964	0.922–1.009
Hb (g/dL)	−0.008	0.161	0.003	1	0.959	0.992	0.724–1.359
Albumin (g/dL)	−0.622	0.807	0.594	1	0.441	0.537	0.111–2.610
Calcium (mg/dL)	−1.115	0.594	3.525	1	0.060	0.328	0.102–1.050
Sodium (mg/day)	0.001	0.000	4.625	1	0.032 *	1.001	1.000–1.001
MUFA (g/day)	0.050	0.075	0.445	1	0.505	1.051	0.908–1.218
n-6 PUFA (g/day)	0.000	0.000	0.277	1	0.598	1.000	1.000–1.000
Hypertension	0.828	0.335	6.100	1	0.014 *	2.288	1.186–4.412

BFP: body fat percentage, MUFA: monounsaturated fatty acid, PUFA: polyunsaturated fatty acid, *: *p* < 0.05. Covariates: age, BFP, grip strength, hemoglobin, albumin, calcium, sodium, MUFA, n-6 PUFA, and hypertension. There were significant differences in BFP, sodium, and hypertension.

## Data Availability

The data of health check-ups used to support the findings of this study are available from the corresponding author upon request.

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
