# Peer review of "Nutritional Influences on Locomotive Syndrome"

_jcm, 2022, doi:10.3390/jcm11030610_

Round 1

Reviewer 1 Report

Although the manuscript certainly has merits, there are a couple of things that I see as needing improvement before publication:

 - There is a need to do another correction for language - as the grammar is not always used correctly;

 - The results are difficult to read - presenting everything in tables makes understanding differences between groups hard to spot and follow. Another graphic representation should be used for some of the items or the data in the tables should be limited. 

- The number of citations is high -the authors cite two articles to demonstrate the same affirmation in several parts of the research. As some of these things are already recognized in the medical community, reducing the number of citations by just citing one article, is appropriate as it would make the verification of some of the pieces of information easier. 

Author Response

Comments and Suggestions for Authors

Although the manuscript certainly has merits, there are a couple of things that I see as needing improvement before publication:

 - There is a need to do another correction for language - as the grammar is not always used correctly;

Thank you for your comments.

We had our manuscript proofread again by an English editing expert.

 - The results are difficult to read - presenting everything in tables makes understanding differences between groups hard to spot and follow. Another graphic representation should be used for some of the items or the data in the tables should be limited. 

Tables 1, 2, and 4 were changed to only include items with significant differences and relevant items. In addition, we attached a table with all the parameters as supplementary data.

- The number of citations is high -the authors cite two articles to demonstrate the same affirmation in several parts of the research. As some of these things are already recognized in the medical community, reducing the number of citations by just citing one article, is appropriate as it would make the verification of some of the pieces of information easier. 

We reduced the number of citations in multiple references to a single reference.

Reviewer 2 Report

jcm-1528121

Thank you for the opportunity to review this manuscript. The authors studied association betwaeen locomotive syndrome and nutritional intake. Overall, only minor comments.

ABSTRACT:

Line 14: LDL-C should be written out.

INTRODUCTION:

Line 34: is there text missing? Or just no capitalize letter?

METHODS:

Line 139: Please define the metabolic parameters examined. What were the CV?

Line 158: Please provide more detail on how the dietary data was analyzed.

Line 171: please provide more details and explanations for the statistical analysis. What were covariates? Where there any?

DISCUSSION

Line 327: BIA is not the gold standard assessment for body composition (4 compartment model is, or DXA). Please include.

TABLES

Clearly legends are needed on all tables. Especially 4 and 5 where non are provided.

Table 1 and 2: Platelet units are odd. Mean and SD? Median and IQR? N (%)? How is data presented in this table? Avoid abbreviations in the left-hand column as much as possible.

Author Response

Comments and Suggestions for Authors

jcm-1528121

Thank you for the opportunity to review this manuscript. The authors studied association betwaeen locomotive syndrome and nutritional intake. Overall, only minor comments.

ABSTRACT:

Line 14: LDL-C should be written out.

I added Low Density Lipoprotein Cholesterol in abstract.

INTRODUCTION:

Line 34: is there text missing? Or just no capitalize letter?

 I changed as follows;

Visceral diseases and genetic factors can also affect LS.

METHODS:

Line 139: Please define the metabolic parameters examined. What were the CV?

I added in Blood sample assessment as follows

2.5 Blood sample assessment

At the checkup, fasting blood samples were collected by venipuncture and centrifuged within 1 hour of collection. Serum samples were stored at -80°C until measurements were taken. Routine biochemical analysis was performed in the laboratory of Yakumo Town Hospital [22]. The following items were investigated; White blood cell,  Hemoglobin, Platelet,  HbA1c, Total Protein, Serum Albumin, alkaline phosphatase (ALP), aspartate transaminase (AST), alanine aminotransferase (ALT),γ-glutamyltranspeptidase, Total-cholesterol, Triglyceride, High density Lipoprotein Cholesterol( (HDL-C), Low Density Lipoprotein Cholesterol (LDL-C), Blood urea nitrogen, Creatinine, Uric asid, Calcium, C-reactive protein.

Line 158: Please provide more detail on how the dietary data was analyzed.

I added the explanation of FFQ.

Line 171: please provide more details and explanations for the statistical analysis. What were covariates? Where there any?

The parameters that showed significant differences in the univariate analyses were used as covariates in the logistic regression analysis.

I changed statistical analysis as follows;

2.8 Statistical analyses

We divided all participants into adults(< 65 years old) and older adults(≥ 65 years old). Continuous variables are expressed as mean ± standard deviation (SD). We compared continuous variables of the L group to those of the N group using the Student’s t-test and categorical variables using the chi-square test. These analyses were conducted for total, adult and older adult respectively. Statistical significance was set at p <0.05. All the parameters listed in the supplement tables were examined. In addition, the table shows the parameters that showed significant differences and their related parameters.

Logistic regression analysis was performed on each adults and older adults to evaluate the important risk factors in the L group. In the logistic regression analyses, we defined that the dependent variable was the group and that the covariables were the parameters that showed significant differences in the Student t-test and the chi-square test comparing L Group and N Group. All statistical analyses were performed using SPSS Statistics v.28.0 software for Mac (IBM Corp., Armonk, NY, USA). Statistical significance was set at p <0.05.

DISCUSSION

Line 327: BIA is not the gold standard assessment for body composition (4 compartment model is, or DXA). Please include.

 I added this sentence in the limitations.

TABLES

Clearly legends are needed on all tables. Especially 4 and 5 where non are provided.

I added the legends in all tables.

Table 1 and 2: Platelet units are odd. Mean and SD? Median and IQR? N (%)? How is data presented in this table? Avoid abbreviations in the left-hand column as much as possible.

I corrected Platelet units.

I added “Valuse are expressed as means ± standard deviations.” in table legends.

I changed items that do not explain abbreviations to complete expressions.

Reviewer 3 Report

MS #: jcm-1528121

The comments on this paper

Title: Nutritional influences on locomotive syndrome

The authors performed the cross-sectional study to examine the relationship between locomotive syndrome (LS) and nutritional intake, and identify the nutrients and dietary habits that influence on LS. In conclusion, Salt intake and hypertension were associated with the LS in this study. This study was well designed and evaluated properly, and the paper is written logically. However, there are several concerns on this paper before publication.   

General concerns

  1. In this study, the nutritional intake status was evaluated using the questionnaire (FFQ). The conclusion, that salt intake was associated with LS, was based on the results from this questionnaire. How long does the result of this questionnaire reflect nutritional intake? Was the study performed once a year? When (Which season) was the study being held? The season, which was winter or summer, might influence on the results of this questionnaire? A more detail explanation of this questionnaire might be needed to understand the relationship between the nutritional status and LS.
  2. The definition of exclusion criteria. Was diabetes excluded in this study? Why does the incidence of diabetes describe in the results?

Specific comments

  1. Introduction, line 34. “visceral diseases” should be “Visceral diseases”.
  2. Materials and Methods. 2.1., line 66. Diabetes was included in the exclusion criteria. However, Table 1 shows that 24 subjects were diabetes in this study. What was the definition of exclusion criteria in this study?
  3. Materials and Methods. 2.3., line 82. “The stages” should be “the stages”?
  4. Materials and Methods. 2.4., line 124. The space might be needed between age and (BFP).
  5. Materials and Methods. 2.6., lines 149-151. Did the subjects with diabetes excluded or not?
  6. 3.2., line 223. Table 1?? Should it be Table 4?
  7. Results. 3.2, line 233 and 234. The results of sodium and n-6 PUFA, the space should be needed before and after ±. 

Author Response

Comments and Suggestions for Authors

MS #: jcm-1528121

The comments on this paper

Title: Nutritional influences on locomotive syndrome

The authors performed the cross-sectional study to examine the relationship between locomotive syndrome (LS) and nutritional intake, and identify the nutrients and dietary habits that influence on LS. In conclusion, Salt intake and hypertension were associated with the LS in this study. This study was well designed and evaluated properly, and the paper is written logically. However, there are several concerns on this paper before publication.   

General concerns

  1. In this study, the nutritional intake status was evaluated using the questionnaire (FFQ). The conclusion, that salt intake was associated with LS, was based on the results from this questionnaire. How long does the result of this questionnaire reflect nutritional intake? Was the study performed once a year? When (Which season) was the study being held? The season, which was winter or summer, might influence on the results of this questionnaire? A more detail explanation of this questionnaire might be needed to understand the relationship between the nutritional status and LS.

The explanation of FFQ was added.

The season for this checkup is summer.

However, since the FFQ is a survey looking back over the past year, we considered the survey to be irrelevant to the season itself.

  1. The definition of exclusion criteria. Was diabetes excluded in this study? Why does the incidence of diabetes describe in the results?

Patients undergoing treatment for diabetes mellitus were excluded.

Participants whose blood tests met the definition of diabetes as defined in this study were considered diabetes mellitus.

I changed the exclusion criteria as follows;

2.1. Study participants

The individuals surveyed were volunteers who underwent a municipal-supported health checkup in Yakumo in 2016 and 2017. Data from 2016 were used for those who participated in two consecutive years (2016 and 2017). Yakumo has a population of approximately 17,000, of whom 28% are >65 years old. More people in this town engage in agriculture and fishing than those in urban areas. Yakumo has conducted annual health checkups since 1982. Physical examinations included voluntary orthopedic and physical function tests and internal examinations. Psychological examinations and a health-related QOL survey (SF-36) were also conducted [14]. This study included all participants who completed an assessment of the LS risk stage, bioelectrical impedance analysis (BIA), fasting blood samples, and nutritional intake status. The exclusion criteria were as follows: history of spine or joint surgery, severe knee injury, severe hip osteoarthritis, history of hip or spine fractures, neuropathy, severe mental illness, diabetes that was diagnosed and treated by a physician, kidney or heart disease, non-fasting, severe impairment of walking or standing, and impairment of the central or peripheral nervous system.

Specific comments

  1. Introduction, line 34. “visceral diseases” should be “Visceral diseases”.

I corrected it.

  1. Materials and Methods. 2.1., line 66. Diabetes was included in the exclusion criteria. However, Table 1 shows that 24 subjects were diabetes in this study. What was the definition of exclusion criteria in this study?

Patients undergoing treatment for diabetes mellitus were excluded.

Participants whose blood tests met the definition of diabetes as defined in this study were considered diabetes mellitus.

  1. Materials and Methods. 2.3., line 82. “The stages” should be “the stages”?

Thank you for your comments.

The sentence pointed out by you was changed due to re-editing by the expert.

  1. Materials and Methods. 2.4., line 124. The space might be needed between age and (BFP).

I added space.

  1. Materials and Methods. 2.6., lines 149-151. Did the subjects with diabetes excluded or not?

Patients undergoing treatment for diabetes mellitus were excluded.

  1. 3.2., line 223. Table 1?? Should it be Table 4?

I corrected it.

  1. Results. 3.2, line 233 and 234. The results of sodium and n-6 PUFA, the space should be needed before and after ±. T.

I corrected it.
